# Approaching the Gut and Nasal Microbiota in Parkinson’s Disease in the Era of the Seed Amplification Assays

**DOI:** 10.3390/brainsci12111579

**Published:** 2022-11-19

**Authors:** Alessandra Consonni, Martina Miglietti, Chiara Maria Giulia De Luca, Federico Angelo Cazzaniga, Arianna Ciullini, Ilaria Linda Dellarole, Giuseppe Bufano, Alessio Di Fonzo, Giorgio Giaccone, Fulvio Baggi, Fabio Moda

**Affiliations:** 1Division of Neurology 4-Neuroimmunology and Neuromuscular Diseases, Fondazione IRCCS Istituto Neurologico Carlo Besta, 20133 Milan, Italy; 2Division of Neurology 5-Neuropathology, Fondazione IRCCS Istituto Neurologico Carlo Besta, 20133 Milan, Italy; 3Division of Neurology, Fondazione IRCCS Ca’ Granda Ospedale Maggiore Policlinico, 20122 Milan, Italy

**Keywords:** Parkinson’s disease, microbiota, α-synuclein, olfactory mucosa, seed amplification assays

## Abstract

Parkinson’s disease (PD) is a neurodegenerative disorder often associated with pre-motor symptoms involving both gastrointestinal and olfactory tissues. PD patients frequently suffer from hyposmia, hyposalivation, dysphagia and gastrointestinal dysfunctions. During the last few years it has been speculated that microbial agents could play a crucial role in PD. In particular, alterations of the microbiota composition (dysbiosis) might contribute to the formation of misfolded α-synuclein, which is believed to be the leading cause of PD. However, while several findings confirmed that there might be an important link between intestinal microbiota alterations and PD onset, little is known about the potential contribution of the nasal microbiota. Here, we describe the latest findings on this topic by considering that more than 80% of patients with PD develop remarkable olfactory deficits in their prodromal disease stage. Therefore, the nasal microbiota might contribute to PD, eventually boosting the gut microbiota in promoting disease onset. Finally, we present the applications of the seed amplification assays to the study of the gut and olfactory mucosa of PD patients, and how they could be exploited to investigate whether pathogenic bacteria present in the gut and the nose might promote α-synuclein misfolding and aggregation.

## 1. Introduction

The human body hosts and interacts with a huge variety of microorganisms, which are mostly harmless and sometimes essential. They are referred to as microbiota and comprise bacteria, archaea, eukaryotes and viruses and colonize several parts of the body including the skin and different mucosa of the urogenital, gastrointestinal and respiratory tract [1]. The term microbiota defines the complex population of microorganisms located in specific ecological niches, while the entire genetic heritage is referred to as the microbiome; although the two terms have different meanings, microbiota and microbiome are often used interchangeably by the scientific community [2]. The microbiota has a crucial role in governing pathogenic microorganisms in a symbiotic state and in maintaining the balance of the total bacterial population, a condition referred to as eubiosis. Perturbation of the normal microbiota composition and diversity (dysbiosis) allows the growth of pathogenic populations [3,4] and has been associated with several diseases, including inflammatory bowel disease [5], inflammatory arthritis [6], different types of cancers [7], and also neurological conditions [8]. In recent years, large-scale metagenomics projects, e.g., the Human Microbiome Project, placed the microbiota under the spotlight of research on its role in the health and pathogenesis of several diseases. Indeed, thanks to the increasing availability of high-throughput DNA sequencing techniques (e.g., Next Generation Sequencing, NGS), it is possible to isolate and sequence microbial DNA to characterize the microbiota composition in human biological samples [9]. Two techniques have been used to study the microbial community: NGS sequencing of the 16S ribosomal RNA gene variable regions (for meta-taxonomy), or whole-metagenome shotgun NGS sequencing (for functional gene composition analysis). Sequencing of the 16S rRNA gene, encoding for the 16S subunit protein (a component of the 30S subunit of a prokaryotic ribosome), enables the identification, classification and relative quantification of the different bacterial species present in the sample [10]. The 16s rRNA gene is a 1.5 kbp gene containing highly conserved regions as well as nine hypervariable regions (V1-V9), whose analysis generates a ‘fingerprint’ of individual species or closely related species. The degree of conservation varies widely between hypervariable regions, with more conserved regions related to higher-level taxonomy and less conserved regions to lower levels, such as genus and species. Sequences resulting from the NGS analysis can be compared to reference genomes and/or clustered de novo to identify the bacterial species present in the sample. This approach is used to describe the microbial community in combination with quantitative measures such as alpha and beta diversity, species evenness, and the relative abundance of particular groups of closely related species (operational taxonomic units, OTU) [11,12,13]. Several studies have shown that alterations of the intestinal microbiota can also affect the physiology of the central nervous system (CNS). Indeed, thanks to the “gut-microbiota-brain axis”, the enteric nervous system (ENS) and the CNS are linked to each other and can communicate in a bidirectional way. Perturbations of the gut microbiota have been included among the possible causative agents of several neurodegenerative diseases, such as Alzheimer’s disease (AD), multiple sclerosis (MS) and Parkinson’s disease (PD) [8]. In particular, several groups have demonstrated that changes in the gut bacterial community can trigger alpha-synuclein (α-syn) pathology in PD. With the development of the seed amplification assays (SAAs), it is now possible to detect trace amounts of disease-associated α-syn (α-syn^D^) in several peripheral tissues (e.g., the gut) of PD patients or other α-synucleinopathies, including multiple system atrophy (MSA) and dementia with Lewy bodies (DLB) [14]. In this review, we will discuss the most recent evidence, which describes possible links between gut/nasal dysbiosis and PD, and the potential contribution that pathogenic bacteria might have on α-syn^D^ generation in the era of the SAAs.

## 2. Gut Microbiota

Most of the microorganisms that constitute the human microbiota resides in the gut and, due to the stability/resilience and the symbiotic interaction with the host, it is possible to consider this huge community of cells a real organ. It has been estimated that the human gastrointestinal (GI) tract contains more than 100 trillion microorganisms, including bacteria, yeast and viruses, and the gut microbiota encodes over 3 million genes producing thousands of metabolites. Each individual is provided with a unique, complex and dynamic microorganism profile that plays many specific functions in host nutrient metabolism, maintenance of the structural integrity of the gut mucosal barrier, immunomodulation, and protection against pathogens [15]. Gut microbiota is composed of different bacteria species taxonomically classified by species, genus, family, order, class and phyla. The dominant microbial phyla in the gut are Firmicutes, Bacteroidetes, Actinobacteria, Proteobacteria, Verrucomicrobia and Fusobacteria, with Firmicutes and Bacteroidetes representing 90% of the total gut microorganisms [16]. At least 200 different genera are part of the Firmicutes phylum and, among them, the most representative are: *Lactobacillus* spp., *Bacillus* spp., *Clostridium* spp., *Enterococcus* spp. and *Ruminicoccus* spp. Bacteroidetes consist of predominant genera such as *Bacteroides* and *Prevotella*, while the Actinobacteria phylum is proportionally less abundant and mainly represented by the *Bifidobacterium* genus [17]. Human gut microbiota composition differs taxonomically and functionally in different GI regions, which are markedly distinct in terms of physiology, digestion flow rates, substrate availability, host secretions, pH and oxygen tension [18]. Indeed, the small intestine, due to the short transit times (3–5 h) and the high bile concentrations, is a challenging environment for microbes, while the large intestine has a great microbial community, being dominated by obligate anaerobes [19]. Moreover, microbiota composition varies according to the life stage (pre-natal, newborn, first three years of age); it stabilizes during the adult period, undergoing variations due to environmental or health factors. From childhood to old age, where there is often observed a progressive increase in Firmicutes/Bacteroidetes ratio, different factors can influence the diversity of the microbiota, such as gender, body mass index, type of diet, geographical area of living, the presence of antibacterial agents or pre/probiotics, treatment with specific drugs (e.g., antibiotics), and pathological conditions [20,21,22,23]. As previously mentioned, the balance between the bacterial species composing the microbiota is referred to as eubiosis; the eubiosis/dysbiosis condition strongly influences our health and disease status. Healthy intestinal flora is essential to promote the health of the host, but the excessive growth of the bacterial population leads to a variety of harmful conditions. Several epidemiologic studies demonstrated that various gastrointestinal disorders (e.g., inflammatory bowel disease), metabolic diseases (e.g., diabetes) and autoimmune diseases (e.g., celiac disease) can alter the microbial balance, thus promoting dysbiosis with a consequent modification of the distribution and the composition of the microbiota [24,25,26,27]. In particular, an alteration of the balance between symbiotic bacteria (*Lactobacillus* spp., *Bifidobacterium* spp., *Enterococcus* spp., *Propionibacterium* spp. and *Peptostreptococcus* spp.) and opportunists (*Bacteroides*, *Bacilli*, *Clostridia*, *Enterobacteriales*, *Peptococcus* spp., *Staphylococcus* spp. as well as yeasts) is associated with gastrointestinal disorders. In general, the symbiotic relationship between the gut microbiota and the host is regulated and stabilized by a complex network of interactions that encompass metabolic, immune and neuroendocrine crosstalk. This crosstalk is potentially mediated by microbial-synthesized metabolites which exhibit pleiotropic effects, for instance by acting as signalling molecules in regulating host neuro-immune-inflammatory axes that could physiologically link the gut with other organ systems.

### The Gut-Brain Axis: Gut Microbiota and Central Nervous System Interaction

In recent years, it became evident that intestinal bacteria could alter CNS physiology by promoting pro-inflammatory mechanisms, and that dysbiosis could contribute to pathological processes in complex neurological conditions [28]. The communication between gut microbiota and CNS develops through the so-called gut-brain axis (GBA), which consists of a bidirectional language between CNS and ENS, linking emotional and cognitive centres of the brain with peripheral intestinal functions. Recent advances in research have described the importance of gut microbiota in influencing these interactions [29,30]. The CNS can modulate the motility, the permeability and the secretion of intestinal mucus through the autonomic nervous system, or activate the hypothalamic-pituitary-adrenal axis and induce overproduction of cortisol, with a consequent reduction of the expression of adhesion proteins in the intestinal epithelium, and an increase of its permeability [31]. In parallel, the intestinal microbiota can deliver signals (including specific neurotransmitters and metabolites) that can reach the CNS either through the ENS or through the blood flow, also considering their ability to cross the blood brain barrier (BBB) [32,33]. The CNS has long been considered an immune-privileged organ, with the BBB strictly regulating the movements of molecules, ions and cells from the periphery via tight junctions [34]. However, this concept has been reconsidered, as compelling evidence suggests that immune cells can enter the CNS via BBB, the choroid plexus and the lymphatic vessels, especially in patients with neurological disorders [35]. Bacterial products, metabolites and neurotransmitters, in particular, are described among the molecules that can cross the BBB, promoting not only CNS development but also the onset of certain diseases [36]. Among the bacterial neuroactive molecules functioning as neurotransmitters that directly contribute to brain-gut communication the most relevant are: 1. gamma-aminobutyric acid (GABA), which is synthesized by *Lactobacillus* spp. and *Bifidobacterium* spp. [37]; 2. dopamine and noradrenaline produced by bacterial belonging to *Bacillus* species and *E. coli* [38]; and 3. serotonin, mainly released by *Lactobacillus* spp., *Streptococcus* spp. and *Klebsiella* spp. [39]. Interestingly, 90% of serotonin required for several CNS functions is produced in the gut, and the binding of serotonin to 5-HT receptors on microglial cells promotes the release of cytokine-carrying exosomes, thus providing an important mechanism for the gut-induced modulation of neuroinflammation [40]. Among bacterial metabolites, the tryptophan derivatives and the short-chain fatty acids (SCFAs) play important roles in GBA communication. Metabolites derived from dietary tryptophan could control CNS inflammation, acting on microglia activation and the transcriptional program of astrocytes, thus maintaining brain homeostasis [41]. SCFAs are the main metabolites produced by bacterial fermentation of dietary fibres in the colon; SCFAs can bind to specific immunological receptors and, subsequently, be used as sources of energy, and gene expression regulators of epithelial integrity and immune interactions [42]. The three main components of SCFAs (acetate, propionate and butyrate) are also involved in specific neurological functions, such as: the stimulation of enterochromaffin cells for the production of serotonin; the control of the maturation and development of microglia; the modulation of nerve growth factor (NGF) and glial cell-derived neurotrophic factor (GDNF). These latter are important mediators of neuronal growth, survival and differentiation, and are also necessary for the formation of the synapses [43]. In particular, acetate alters the levels of the neurotransmitters glutamate, glutamine and GABA in the hypothalamus, while propionate and butyrate act on the neurochemical balance of the brain, by regulating the expression levels of tryptophan 5-hydroxylase 1, an enzyme implicated in serotonin synthesis, and of tyrosine hydroxylase, involved in the biosynthesis of dopamine, norepinephrine and epinephrine [44]. Starting from this evidence, which highlights the close communication between the CNS and gut, recent studies correlated the dysbiosis of the gut microbiota with behavioural and neurological pathologies such as depression, the disorder of autism spectrum (ASD), Alzheimer’s disease (AD), multiple sclerosis (MS), amyotrophic lateral sclerosis (ALS), prion diseases and PD [45,46,47,48,49,50]. In the next paragraphs of this review, we summarize the available evidence and new potential mechanisms linking gut and nasal microbiota dysbiosis with the onset and progression of PD.

## 3. Nasal Microbiota

The interconnection between gut and oral/nasal microbiota is still poorly described. However, it is known that nasal microbiota can influence the olfaction, the immune regulation, and the homeostasis of the CNS. The respiratory tract is a complex system in which microbiota acts as a “guardian” that provides resistance to colonization by pathogens, and the nasal region is one of the distinct environments for the survival of specific microorganisms. The upper respiratory tract (URT) is mainly composed of distinct anatomical structures, with different cell populations differently exposed to external factors [51]. Dissimilarities in the microbiota composition at lower taxonomic levels in this region are found, due to the specific influences from the external (e.g., humidity, temperature, relative oxygen concentration), as well as from interactions between microbes and the immune system. Colonization of the URT begins at birth and, during early life, respiratory bacterial communities are highly dynamic and mainly influenced by the mode of birth, feeding type, crowding conditions and antibiotic treatments. Throughout adulthood, the nasal microbiota stabilizes and remains mainly constant with a stable composition that hampers pathogen overgrowth, or may develop an unstable bacterial community which predisposes to infection and inflammation [52,53]. Considering the URT structure (nasal cavity, nasopharynx and oropharynx), the nasal cavity is closely linked to the external environment, and is dominated by *Staphylococcus* spp., *Propionibacterium* spp., *Corynebacterium* spp., *Moraxella* spp., *Dolosigranum* spp., and *Streptococcus* spp. [54]. The nasopharynx, is mainly composed of microbial communities typically characterized by *Moraxella* spp., *Staphylococcus* spp., and *Corynebacterium* spp. The oropharynx has a more complex microbiota that includes *Neisseria* spp., *Rothia* spp., and anaerobic bacteria among which *Veillonella* spp., *Prevotella* spp., and *Leptotrichia* spp. As previously reported, the primary function of any microbial ecosystem is to maintain a state of symbiosis, providing resistance against pathogens. In this context, the URT is considered as a major reservoir of potential pathogens, e.g., Streptococcus pneumoniae and Haemophilus influenzae. The nasal eubiosis is necessary to limit the expansion and subsequently spreading of these microorganisms towards the lungs, potentially leading to symptomatic infection. Thus, establishing and maintaining a balanced microbiota in the URT resilient to pathogenic expansion and invasion is vital for respiratory health [55,56]. The epithelial layer of the nasal cavity acts as a barrier, thus protecting from pathogen invasion and preventing pathogens from reaching the lower respiratory tract (e.g., the lungs). The epithelial cells are involved in immune defence mechanisms through the secretion of lysozyme, lactoferrin, IgM, and IgA, keeping a healthy nasal environment and preventing local infections and inflammations [57,58]. Some evidence suggests that nasal microbiota and its metabolites can reach the CNS through the olfactory system (involving the olfactory bulb) by escaping the BBB [59]. 

### The Nose-Brain Axis: Nasal Microbiota and Central Nervous System Interaction 

The gut-brain axis is an essential bidirectional communication way that links two important organs and that influence their physiological or pathological conditions. In 2008, Doty and colleagues first proposed the nose-to-brain hypothesis [60]. This hypothesis describes the olfactory system as a portal from which environmental factors, including microorganisms, can access the olfactory bulb (OB) and directly spread to the brain. For this reason, the olfactory system may be involved in the onset of some neurological disorders. There are two main mechanisms by which nasal microbes might contribute to the initiation and progression of neurological diseases. First, bacteria or their products could directly spread from the nose to the CNS along neuronal pathways promoting inflammation, the accumulation of pathological proteins such as α-syn or amyloid beta (Aβ), and finally neurodegeneration. In AD and PD, in particular, there could be a prion-like spread of misfolded proteins from the olfactory system through neuronal connections [61]. Moreover, it has been described that an injection of α-syn or Aβ oligomers into the OB of mice are rapidly transferred to the interconnected brain regions [62,63]. Similarly, inflammation triggered by nasal dysbiosis could also propagate throughout the CNS, and OB microglia have an important role in this process. Injection of a single dose of lipopolysaccharide (LPS) into the mouse nose leads to a wave of microglial toll-like receptor 2 (TLR2) induction, which begins in the OB and progresses throughout connected regions of the brain. This activation triggers the production of pro-inflammatory cytokines, resulting in neuronal and oligodendrocyte damage [64]. The second mechanism through which nasal microbiota could influence the development of neurological disease is through the lymphatic system, in particular, via drainage of the posterior nasal sinuses into the cerebrospinal fluid (CSF), considered a route by which bacterial toxins could enter the CNS [65]. A recent 16S rRNA NGS analysis shows that infection with Chlamydia pneumonia, a respiratory pathogen, is associated with an increased occurrence of AD. C. pneumoniae DNA is found throughout the olfactory system and in brain tissue samples of post-mortem AD patients, while it lacks in controls [66]. To corroborate these data, it has been described that intranasal inoculation of C. pneumoniae in animal models (BALB/c mice) results in the progressive infection of neuronal cells with this microorganism and promotes Aβ accumulation throughout the brain [67]. Although C. pneumoniae is certainly a pathogenic species, it is present in the nasal microbiota in low abundance. Dysbiosis could therefore allow amplification of this C. pneumoniae population and lead to olfactory nerve infiltration and access to the CNS [67]. In a recent paper, Hoggard and colleagues hypothesized that inflammatory conditions driven by altered nasal microbiota in the URT of patients with chronic rhinosinusitis (CRS) could have a potential role as a modifier of neural signalling leading to mental health impairment [68]. In this study, the authors investigated associations between the nasal microbiota, the local concentrations of different neurotransmitters (e.g., serotonin, dopamine, GABA) and the depression severity in a cohort of CRS patients and healthy controls [68]. Interestingly, they found that several commonly “health-associated” bacterial taxa were positively associated with higher neurotransmitter concentrations and negatively associated with depression severity. In contrast, taxa commonly associated with a nasal microbiota dysbiosis negatively associated with neurotransmitters and positively with depression severity. Taken together, the findings obtained from the literature lend support to the potential for downstream effects of the nasal microbiota on neural signalling and, subsequently, brain functions and dysfunctions.

## 4. Parkinson’s Disease

Parkinson’s disease (PD) is a neurodegenerative disease affecting about 1/100 of people in the sixth decade of life. PD is characterized by motor deficits, including slowness of movement, stiffness and tremor [69]. However, before the appearance of these symptoms, it presents a prodromal phase, which is instead characterized by the appearance of non-motor symptoms, which include sleep disturbances (e.g., idiopathic REM Sleep Behaviour Disorder-iRBD), constipation, depression and olfactory impairment [70]. One of the typical neuropathological features of PD concerns the loss of dopaminergic neurons of the substantia nigra pars compacta [71]. The loss of these cells, and the consequent reduction of dopamine release, leads to a dysfunction of other cerebral structures, including the basal ganglia, essential for the initiation and the control of movement. Dopaminergic neurons show the presence of typical protein aggregates, both in the cytoplasm (Lewy bodies) and in neuronal processes (Lewy neurites), mainly composed of misfolded α-syn (α-syn^D^). Lewy bodies are also found in other cerebral regions, such as the dorsal motor nuclei of the vagus nerve, the locus coeruleus and the entorhinal cortex, but the exact trigger for the formation of α-syn^D^ and the consequent neuronal degeneration is not yet clear [72]. The study of family cases led to the identification of genetic risk factors, some with high penetrance but very rare (e.g., mutations in the SNCA gene that encodes for the α-syn protein), while others are more frequent in the population but with reduced penetrance (e.g., variants of the GBA and LRRK2 genes) [73]. Growing evidence suggests that α-syn^D^, in addition to forming intraneuronal protein aggregates which are characteristic of PD, may be directly involved in the onset and progression of the disease. Aggregates of α-syn^D^ propagate between neurons, and they can interact with the normal α-syn (which abounds at the level of the synapses) inducing a conformational change that promotes the formation of new α-syn^D^ species [14,74] that finally aggregate and form the typical protein deposits. Through this mechanism, α-syn^D^ propagates very efficiently within the CNS. Interestingly, in addition to being present in neurons of the CNS, α-syn^D^ was also identified in the ENS, especially in the early stages of PD and it could contribute to the onset of gastrointestinal disorders, including constipation and impaired motor function of the colon [75]. These findings prompted researchers to hypothesize that PD may originate in the gut where alterations of the local microbiota can promote the local formation of α-syn^D^. This pathological protein can finally reach the brain, through the gut-brain axis, causing the clinical and neuropathological changes typical of the disease [76]. For this reason, the correlation studies between intestinal microbiota and PD have been intensified in the last years. Interesting discoveries have shown that, before aggregating in the CNS, α-syn^D^ accumulates in peripheral structures such as the enteric plexus of the stomach and in the OB of PD patients. For this reason, it has been proposed that the formation of α-syn^D^ may occur in peripheral regions (e.g., the gut and nose) and that from those organs it could reach the brain (dual-hit hypothesis) [77]. In the next paragraphs, we will describe the most recent discoveries which link alterations of gut and nasal microbiota with PD onset and progression.

### 4.1. Neuroinflammation in PD

Neuroinflammation plays a crucial role in PD. In particular, it has been shown that inflammation-derived oxidative stress and cytokine-dependent toxicity may contribute to the degeneration of the nigrostriatal pathway and accelerate the progression of the disease [78]. It has been hypothesized that transient initiation factors, like bacterial or viral infection, neuronal injury (e.g., brain trauma), environmental toxins (e.g., pesticides), may induce an increased production of chemokines, cytokines, reactive oxygen species (ROS) and other factors, which are characteristic of chronic neuroinflammation, leading to the activation of glial cells (both astrocytes and microglia) around dopaminergic neurons, thus contributing to neuronal dysfunction and death [79]. In addition, dying dopaminergic neurons release chemoattractants that promote further in situ migration of activated microglia to remove neuronal debris [78]. Interestingly, midbrain dopaminergic neurons exhibit a peculiar sensitivity to inflammatory factors (e.g., TNF) [80]. Moreover, these pro-inflammatory factors are elevated in the CSF and brain of PD patients [81]. Positron emission tomography (PET) imaging studies of microglia activation showed that pons, basal ganglia, striatum and frontal and temporal cortical regions of patients with idiopathic PD have markedly elevated neuroinflammation, compared to healthy subjects [82]. It was also observed that the R47H variant of the triggering receptor expressed on myeloid cells 2 (TREM-2) represents an important genetic risk factor for PD [83], as well as some variants of the human leukocyte antigen gene (HLA-DRA, which codes for the HLA-DR receptor) specifically expressed on microglia [84]. The substantia nigra of individuals with PD shows upregulation of HLA-DR antigens and the presence of HLA-DR-positive reactive microglia [85]. Studies using mouse models of PD showed that the administration of LPS directly into the substantia nigra of mice overexpressing human α-syn (either wild-type or with the A53T mutation) induced neuroinflammation associated with the death of dopaminergic neurons, other than the accumulation of insoluble aggregates of α-syn^D^ in nigral neurons, thus suggesting that the two mechanisms may be somehow linked [86]. Furthermore, an important microglial activation was observed in several toxin-based PD models (e.g., MPTP) of PD [87,88]. In vitro studies, using primary cultures of murine cortical astrocytes, showed that the presence of inflammatory factors, like IFN-γ and IL-1β, causes the activation of astrocytes and the upregulation of nitric oxide (NO) production [89,90]. Increasing levels of NO can favour α-syn aggregation and accumulation due to a decreased proteasome activity [91]. A very recent study evaluated the scores of two non-specific parameters of inflammation (platelet-to-lymphocyte ratio-PLR, and neutrophil-to-lymphocyte ratio-NLR) by counting the neutrophil and lymphocyte rate in blood samples of PD and multiple system atrophy (MSA) patients or healthy controls and observed that these parameters were higher either in PD and MSA than controls, but in PD they both reached a statistical significance, while in MSA, only NLR was significantly higher than controls [92]. This suggests that PD and MSA have distinct neuroinflammatory patterns. Although many questions remain still unanswered, these evidences support the existence of an important link between inflammation, oxidative stress and PD pathology.

### 4.2. Gut Microbiota and PD

α-syn is not only expressed in the CNS, but also by the neurons of the ENS and has a fundamental role in the release/absorption of neurotransmitters [93]. Pathological aggregates of α-syn^D^ were observed in the biopsies of gastrointestinal tissues of patients with PD [94]. Furthermore, α-syn^D^ was also found in salivary glands, esophagus and stomach, organs potentially involved in the common non-motor symptomatology in PD, such as hypersalivation, dysphagia, delayed gastric emptying and gastroparesis [95]. Studies published by Braak et al. in 2003 [96] and Hawkes et al. in 2007 [77] support the involvement of the GI in the development and progression of PD, suggesting that α-syn^D^ may originate in the gastrointestinal system, reach the CNS via retrograde transport along the neural projections of the ENS, and from there spread to different neuroanatomical regions in a caudal-rostral manner. Apart from the demonstration of the presence of α-syn^D^ aggregates in the vagus nerve [97], studies in experimental animal models of PD have demonstrated that the inoculation of α-syn^D^ in the gastrointestinal tract of mice can induce α-syn^D^ deposition in the CNS [98,99]. The presence of healthy gut microbiota has not only beneficial effects on gastrointestinal structures, but can induce the production of SCFAs and promote the integrity of the BBB through the regulation of the proteins of the tight junctions. A condition of dysbiosis at the microbiota level, on the contrary, may be associated with an increase in the relative abundance of detrimental bacterial species, with a consequent alteration of the intestinal integrity and bacterial production of toxins with inflammatory activity (e.g., LPS) [100]. In particular, it is known that LPS interacts peripherally with the cells of the immune system, thus stimulating the release of pro-inflammatory cytokines; moreover, it has been shown that LPS can induce the formation of α-syn^D^ in experimental models [101]. After the heteromolecular interaction between LPS and α-syn, it was possible to observe the formation of α-syn^D^ with well-defined pathologic structural properties [102]. Further evidence suggests that specific bacterial molecules, in particular proteins with amyloid structures, can promote the formation of α-syn^D^. Indeed, the gut microbiota produces several “amyloid” molecules, including the “curli” protein, expressed in abundance from intestinal strains such as Escherichia coli and the phenol soluble modulins (PSMαs) produced from Staphylococcus aureus. In transgenic mice that overexpress human α-syn, the colonization with E. coli could promote α-syn^D^ formation in the gut and its propagation to the brain, while, in vitro studies showed that the presence of PSMαs catalyzes α-syn aggregation [103,104,105]. Alterations of gut microbiota can lead to other important pathological processes associated with the disease. For instance, intestinal bacteria (especially *Prevotella* spp., *Bacteroides* spp., *Lactobacillus* spp., *Bifidobacterium* spp., *Clostridioides* spp., *Enterococcus* spp. and *Ruminococcus* spp.) contribute to the bioavailability of dopamine, a key neurotransmitter in PD, in the ENS and CNS. Several studies have found that changes in the composition of gut microbiota and dopamine production are linked to the clinical manifestations of PD [102]. In particular, the loss of dopamine in the peripheral nervous system (PNS) and ENS causes gastrointestinal dysfunctions, including delayed gastric emptying and gut dysmotility. While some strains of intestinal bacteria exert negative effects by stimulating further inflammatory responses through the production of endotoxins, others were found to have neuroprotective roles on dopaminergic neurons and prevent the loss of dopamine [106]. 

Through molecular genotyping studies of the microbial species present in the GI tract of PD patients, it has been shown that the relative composition (at different levels) of all major bacterial phyla are altered if compared to healthy subjects. Therefore, changes in gut microbial species could be considered prodromal biomarkers indicating the early onset of the disease [107]. In this regard, increased levels of *Akkermansia muciniphila, Enterobacteriales, Eggerthella *spp.*, Oscillibacter, Escherichia/Shigella, Lachnospiraceae* and *Streptococcus,* and decreased levels of *Roseburia, Coprococcus, Faecalibacterium* and *Eubacterium biforme* were described in PD samples [108]. In addition, increased levels of *Enterococcus *spp.*, Bifidobacterium* spp. and *Ruminococcus* spp. and decreased levels of *Prevotella, Bacteroides* and *Clostridium* were also described [109]. Of note, the results regarding the relative abundance of *Lactobacillus* spp. are still controversial [110]. In conclusion, alterations of gut microbiota can promote intestinal inflammation, LPS secretion and the accumulation of α-syn^D^. These latter can spread to the CNS, through the retrograde vagal pathways of innervation, and contribute to the onset of the typical neurodegenerative alterations of PD. The comprehension of the role of the gut microbiota in PD will help to identify new therapies and optimize methods available to prevent, delay or restore dopaminergic deficits of this disorder.

### 4.3. Nasal Microbiota and PD

Dysfunctions of the olfactory system occur in 75–95% of PD patients in the early disease stages. The nasal cavity is considered a secondary site (in addition to the gut) capable to trigger neuroinflammation in PD and, as mentioned before, it has been hypothesized to serve as a route of pathogen invasion/toxin exposure [60]. Similar to the gut, the nasal cavity is colonized by distinct microbial communities in different regions of the URT. The rostral region of the sinusoidal cavity consists of a specialized epithelial layer in close proximity of the OB, with a stable microbial community that plays a role in olfactory development and the function of smell [111,112]. Changes in the microbiota composition of this region have been associated with a pro-inflammatory profile that can cause several diseases, such as the chronic rhinosinusitis and other respiratory syndromes, which are thought to reduce olfaction [113]. The study of the nasal microbiota and its potential association with PD is still at the dawn, and here we report the few studies published in this context [114,115,116,117]. In their research, Pereira et al., analysed the nasal microbiota of 69 PD patients and 67 healthy controls via 16S NGS sequencing (V1–V3 region). Samples were collected between the septum and the middle turbinate via nasal swab. They identified 553 genera, 177 families, 96 orders, 49 classes and 28 phyla; the most common genus present in PD and control groups was *Corynebacterium* followed by *Propionibacterium, Moraxella, Staphylococcus* and *Burkholderia*. Despite the variety of microorganisms found, it was not possible to detect differences in terms of alpha or beta diversity between PD patients and controls [114]. Heintz-Buschart et al., via NGS sequencing for 16S and 18S rRNA and shotgun metagenomic analysis, have studied the microbiota in nasal wash samples collected from 76 PD patients and 78 healthy controls. This work highlighted a high heterogeneity between samples at all the taxonomic levels, with only one OTU of the genus *Corynebacterium* common to all subjects. However, also in this study, the authors did not find significant differences that could discriminate PD patients from controls. Further statistical analyses, considering 70 anthropometric data, allowed to discover that gender and height of the subjects were able to influence the prokaryotic community. Finally, comparisons between PD patients under treatment with levodopa (L-DOPA) and untreated patients showed a greater relative abundance of *Bacillaceae* only in treated patients [115]. Pal et al. analysed the bacterial community present in the nasal cavity of 30 PD patients and 28 healthy subjects, by the multi-amplicon sequencing of 16S rRNA gene, covering all the hypervariable regions (V1–V9). In this study, nasal sampling was performed at the level of the olfactory fissure using a rigid endoscope. Results of this study were more promising, as it was observed that PD patients have larger amounts of “pro-inflammatory” bacterial species including *Moraxella catarrhalis* and *Ralstonia insidiosa* if compared to controls. Interestingly, patients with increased quantities of *M. catarrhalis* (and other pro-inflammatory bacteria strains) showed more severe PD symptoms and reduced amounts of anti-inflammatory bacteria, including *Blautia wexlerae, Lachnospira pectinoschiza* and *Propionibacterium humerusii* (known to produce SCFA in the gastrointestinal tract). Further analyses performed with machine learning approaches allowed the identification of eight characteristic taxa in PD patients: *Escherichia albertii, Peptoniphilus asaccharolyticus, Staphylococcus aureus, Macrococcus brunensis, Ralstonia insidiosa, Staphylococcus epidermidis, Burkholderia xenovorans* and *Acinetobacter guillouiae*. Finally, the severity of motor symptoms (assessed by the Movement Disorder Society-Unified Parkinson’s scale Disease Rating Scale-MDS-UPDRS) and the olfactory functions were positively correlated with *M. catarrhalis* and *S. epidermidis* and with the commensal bacterial species *P. asaccharolyticus.* The results of this study showed a peculiar pro-inflammatory and dysbiotic environment in the deep nose cavity of PD patients compared to healthy subjects [116]. By sequencing the region V4-V5 of bacterial samples collected from the anterior nasal cavity of 91 patients with PD and 91 healthy controls, Li et al. observed high levels of *Stenotrophomonas, u_Corynebacteriaceae* and *Staphylococcus* spp. in both groups of subjects. However, detailed analyses revealed a higher level of *g_unidentified_Corynebacteriaceae, f_Corynebacteriaceae, and s_Corynebacterium sp* KPL1855 in the PD group compared to healthy controls [117]. In conclusion, olfactory alterations are considered as a prodromal symptom of PD, and the nasal dysbiosis might play a crucial role [118]. In addition, such a dysbiosis could promote the formation of α-syn^D^ which, settling in the neurons of the olfactory mucosa, can contribute to the olfactory disturbance and later initiate its retrograde transport to the olfactory bulb and finally into the CNS.

## 5. SAA Analyses of Olfactory Mucosa and Gut of PD Patients

The seed amplification assays (SAAs) are a group of highly sensitive techniques, including the protein misfolding cyclic amplification (PMCA) and the Real-Time Quaking-Induced Conversion (RT-QuIC), that enabled the detection of trace amounts of α-syn^D^ in CSF [119,120,121,122,123], olfactory mucosa (OM) [118,124,125,126,127], submandibular gland [128,129], skin [130,131,132], saliva [133] and more recently the blood [134] of patients with PD and other α-synucleinopathies, sometimes at prodromal disease stages [14,74]. As previously mentioned, the olfactory functions are impaired in most PD patients and this dysfunction has been reported to occur in the early stages of the disease. For this reason, detecting α-syn^D^ in OM samples collected from patients with olfactory impairment might be of utmost importance for identifying patients at higher risk of developing PD or other α-synucleinopathies. Through SAA analyses, different groups have shown that it is possible to detect α-syn^D^ in OM of PD, MSA and DLB patients with high sensitivity and specificity [118,124,125,126,127], even at early stages [125]. Similarly, GI problems, such as constipation, are considered prodromal symptoms of PD. For this reason, some groups have started to analyse gastrointestinal biopsies by SAAs [129,135,136]. As in the case of OM samples, the results obtained in this context are also very promising and indicated that the SAAs could detect α-syn^D^ in samples collected from PD patients at different disease stages [135]. Now, it would be interesting to investigate whether SAAs results correlate with specific alterations of the OM or gut microbiota. Moreover, since SAAs can mimic in vitro the process of α-syn misfolding and aggregation which occurs in vivo, they can be exploited to study whether and to what extent specific bacteria or their metabolites (e.g., amyloid proteins or LPS), are able to trigger α-syn misfolding in vitro (Figure 1). This would consent to deepen the role of GI and OM dysbiosis in α-syn^D^ formation, PD onset and progression. 

## 6. Treatments for PD 

Although there are no therapies capable to halt the neurodegenerative processes of PD [137], there are several treatments that help to relieve the motor and non-motor symptoms. Regarding the motor symptoms, the pivotal therapy of PD consists in the administration of levodopa, a precursor of dopamine, which acts on the dopaminergic pathway. It is currently considered the gold standard therapy for PD and is administered in the form of tablets associated with inhibitors of the peripheral dopa-decarboxylase enzyme (such as carbidopa or benserazide) [138]. This enzyme converts levodopa into dopamine, therefore, its inhibition hampers the metabolization of the administered levodopa at peripheral level thus allowing it to reach the CNS. The prolonged use of levodopa is however linked to the development of motor complications, which are estimated to affect up to 50% of patients after 5 years of disease [139]. Motor complications are largely due to the short plasma half-life of levodopa and the variability in the gastrointestinal absorption of the drug; for this reason, during the course of the disease there is a tendency to progressively increase the administration of levodopa or to administer other drugs in addition to levodopa, such as entacapone, tolcapone and opicapone. This class of drugs increases levodopa half-life by inhibiting the activity of Catechol-O-metyltransferase (COMT) enzyme (which degrades levodopa) [140]. Another class of drugs used to treat PD are the dopamine agonists (e.g., ropyrinol, pramipexole, rotigotine), which act by mimicking the effect of dopamine at the level of dopaminergic receptors [139]. A further class of drugs are monoamine oxidase B (MAO-B) enzyme inhibitors (e.g., selegiline, rasagiline and safinamide). MAO-B enzyme is a cathalizator of dopamine degradation, so these drugs essentially act by increasing the neurotransmitter’s half-life [141]. In most cases, the motor complications associated with the use of levodopa tend to become increasingly severe; therefore, in the advanced stages of the disease, second-level therapies are evaluated. One of these is the continuous infusion of levodopa by placing a percutaneous endoscopic gastrostomy (PEG) directly in the intestine, which guarantees stable levels of the active principle in the blood. Another treatment is deep brain stimulation (DBS), consisting in the implantation of electrodes at the level of the basal ganglia to stimulate them electrically, regulating their signalling and considerably alleviating the motor symptoms of parkinsonism [142]. Regarding the treatment of non-motor symptoms, it is necessary to use drugs that act on specific conditions. In particular, orthostatic hypotension can be treated with fludrocortisone, which increases salt and water retention, or with midodrine, an α1-adrenergic agonist that acts on the arterial tone. Drugs with antimuscarinic action (e.g., oxybutynin or tolterodine) are indicated for urinary incontinence, since they increase the relaxation of the detrusor muscle of the bladder, while constipation can be treated with osmotic laxatives (e.g., macrogol and lubiprostone) [143]. Antipsychotics such as quetiapine and clozapine are indicated to manage possible behavioural disorders. Finally, the use of cholinesterase enzyme inhibitors such as donepezil and rivastigmine can be considered to treat cognitive disorders in advanced stages of the disease [144]. In the light of new evidence about the possible role of microbiota in PD pathology, some clinical trials have focused on enriching the diet of PD patients with probiotics or prebiotics, or a combination of both. It was found that the consumption of fermented milk, containing multiple probiotic strains and prebiotic fibre, could improve the stool consistency and increase the frequency of complete bowel movements in PD patients with constipation, thus alleviating abdominal pain and the sensation of incomplete emptying [145,146].

Several research groups are actively working to find an effective treatment for PD [147]. In this context, the SAA represents a valuable tool exploitable to monitor disease progression but overall to assess the effects of disease modifying treatments using easy to get biological tissues.

## 7. Conclusions and Future Perspectives

In recent years, research on the microbiota field has taken significant steps forward, mainly thanks to the development of new technologies (e.g., high-throughput DNA sequencing techniques, shotgun metagenomics, metatrascriptomics, metabolomics, metaproteomics), which allow analysing the composition and the functions of the bacterial flora by deepening certain complex aspects that were previously invaluable [148,149]. These analyses have shown that gut dysbiosis could lead to the onset of various pathologies, which involve not only the GI system but, in some cases, also the CNS. In this regard, alterations of the gut microbiota were found to precede the onset of PD and might have a role in α-syn^D^ formation [48]. Another sign that anticipates PD is the loss of smell. Additionally, in this case, nasal dysbiosis might be responsible for the olfactory impairment and considering the direct link between OM and CNS, these alterations might contribute to the onset of PD. Thanks to the development of the SAAs, traces of α-syn^D^ were found in OM and the gut of PD patients, even in the early stages of the disease, and it would be interesting to study whether and to what extent these microbiota alterations might be involved in α-syn^D^ formation. A few studies have demonstrated that the formation of α-syn^D^ may depend on bacterial products (e.g., amyloid proteins including curli, PSMαs or molecules such as LPS) and considering that these molecules are generated as a result of dysbiosis, it would not be entirely surprising that peripheral organs (connected with the CNS) could serve as sites of origin of the disease. Notably, compared to the gut, the OM can be easily collected and the procedure is not invasive and can be repeated over time. This makes the OM an excellent biological tissue that can be subjected to SAA and microbiota analyses with the aim of improving the diagnosis of PD, especially considering the phenotypic heterogeneity of the disease. This variability seems to be associated with different anomalous conformations of α-syn^D^ (strains). It will be therefore important to evaluate whether different alterations of the microbiota can modulate the misfolding process of α-syn and generate different α-syn^D^ strains. Finally, it is of utmost importance to clarify whether the nasal dysbiosis really plays some role in PD onset and progression. If this was the case, it would be possible to identify new therapeutic targets and plan personalized treatments aimed at restoring the nasal eubiosis by reducing harmful bacterial strains (for example by administering selective antibiotics) and by introducing alternative bacterial strains (such as probiotics). Finally, as there are several prodromal symptoms of the disease, such as sleep disturbances (iRDB), constipation and loss of smell, we may select patients for OM collection and SAA and NGS analyses with the aim of studying whether α-syn^D^ or nasal dysbiosis significantly anticipate the onset of PD. This will help to identify patients at a higher risk of developing the disease. Early PD diagnosis and patient stratification is the key to maximize the effectiveness of treatments and to slow or prevent disease progression. SAA is not only useful to recognize and even stratify PD patients in their early disease stages by analysing easily collectible tissues, but can also be exploited to monitor disease progression (especially in patients under pharmacological treatment) and to study the effect of different bacteria on α-syn^D^ generation in vitro. Indeed, since SAA mimics the process of α-synuclein misfolding which occurs in vivo, it represents an important platform to evaluate the influence of harmful bacteria (found in the gut or nose of PD patients) on α-syn^D^ generation and strains formation. This will consent to plan innovative therapeutic strategies by looking at PD from a different perspective. 

## Figures and Tables

**Figure 1 brainsci-12-01579-f001:**
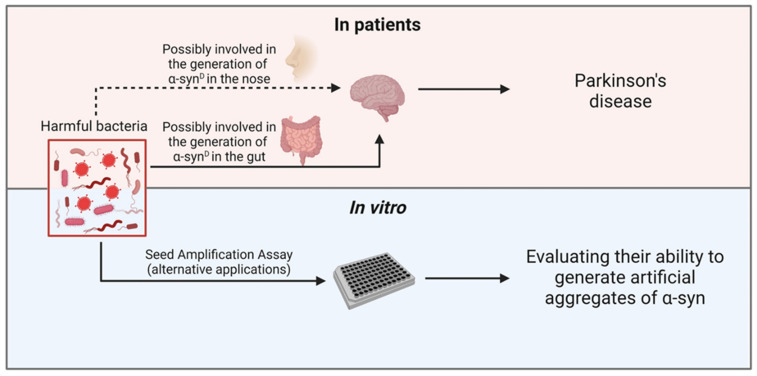
Schematic overview of the principal topics described in the review.

## Data Availability

Not applicable.

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
