# Peer review of "Approaching the Gut and Nasal Microbiota in Parkinson’s Disease in the Era of the Seed Amplification Assays"

_brainsci, 2022, doi:10.3390/brainsci12111579_

Round 1
Reviewer 1 Report
I believe that the paper is relatively well written however the discussion should be enriched. Growing interest is associated with the overlapping of neuroinflammation and neurodegeneration, which might be the case in PD. As gut microbiota is a feature possibly impacting these processes, I assume that extending the discussion by this issue would be valuable.
The understanding role of nasal and gut microbiota in Parkinson's disease is evolving, however the mechanism of this pathway is not fully specified. In this review authors bring an overview, however:
1. I don't see how this review moves us forward as the literature, authors should explain its novelty.
2. It would be valuable to elaborate more extensively on the role of neuroinflammation and microglial activation in the neurodegenerative processes.
Ref.
Reactive microglia are positive for HLA-DR in the substantia nigra of Parkinson's and Alzheimer's disease brains. Neurology. 1988 Aug;38(8):1285-91. doi: 10.1212/wnl.38.8.1285. PMID: 3399080.
Platelet-to-lymphocyte ratio and neutrophil-tolymphocyte ratio may reflect differences in PD and MSA-P neuroinflammation patterns. Neurol Neurochir Pol. 2022;56(2):148-155. doi: 10.5603/PJNNS.a2022.0014. Epub 2022 Feb 4. PMID: 35118638.
3. Perhaps separate paragraphs regarding the possibilities of treatment and limitations should be added.
Reviewer 2 Report
This manuscript is well written for the updated studies in the field of microbiota in PD.
Reviewer 3 Report
Review for brainsci-1929096
The manuscript by Alessandra Consonni et al., entitled “Approaching the gut and nasal microbiota in Parkinson’s disease in the era of the seed amplification assays” is a review article that reviews the crucial role of microbial agents in PD. The authors described the latest findings on microbiota by considering that more than 80% of patients with PD develop remarkable olfactory deficits in their prodromal disease stage. Therefore, the nasal microbiota might contribute to PD, eventually boosting the gut microbiota in promoting disease onset. They presented the applications of the seed amplification assays to the study of the gut and olfactory mucosa of PD patients and how they could be exploited to investigate whether pathogenic bacteria present in the gut and the nose might promote α-synuclein misfolding and aggregation. The reviewed manuscript is well-written, and the cited references are appropriate. The review is important for the PD research field. However, some minor remarks should be taken by authors under consideration. The manuscript needs minor revision.
Comments:
1. If it is possible, authors may include the schematic diagram or graphical diagram for the review. It will easy understanding to the reader and better serve the scientific community.
Round 2
Reviewer 1 Report
I do not have further comments.